# Oxidative Stress and Antioxidant Nanotherapeutic Approaches for Inflammatory Bowel Disease

**DOI:** 10.3390/biomedicines10010085

**Published:** 2021-12-31

**Authors:** Ping Liu, Yixuan Li, Ran Wang, Fazheng Ren, Xiaoyu Wang

**Affiliations:** 1Key Laboratory of Precision Nutrition and Food Quality, Department of Nutrition and Health, China Agricultural University, Beijing 100083, China; ping.liu1@cau.edu.cn (P.L.); liyixuan@cau.edu.cn (Y.L.); wangran@cau.edu.cn (R.W.); renfazheng@263.net (F.R.); 2Key Laboratory of Functional Dairy, Ministry of Education, College of Food Science and Nutritional Engineering, China Agricultural University, Beijing 100083, China

**Keywords:** oxidative stress, reactive species, inflammatory bowel disease (IBD), antioxidant pathways, nano-delivery systems

## Abstract

Oxidative stress, caused by the accumulation of reactive species, is associated with the initiation and progress of inflammatory bowel disease (IBD). The investigation of antioxidants to target overexpressed reactive species and modulate oxidant stress pathways becomes an important therapeutic option. Nowadays, antioxidative nanotechnology has emerged as a novel strategy. The nanocarriers have shown many advantages in comparison with conventional antioxidants, owing to their on-site accumulation, stability of antioxidants, and most importantly, intrinsic multiple reactive species scavenging or catalyzing properties. This review concludes an up-to-date summary of IBD nanomedicines according to the classification of the delivered antioxidants. Moreover, the concerns and future perspectives in this study field are also discussed.

## 1. Introduction

Oxidative stress is the imbalance between the generation of reactive species and the ability to defend against oxidative damage that may lead to the disruption of biological systems. Both oxidation and reduction processes can be generated from endogenous and exogenous sources [1]. It is a cause of a wide range of diseases, including chronic obstructive pulmonary disease, cardiovascular diseases, neurodegenerative diseases, chronic kidney disease, and cancer as well as IBD [2,3]. In intestinal tissues, the response of oxidative stress and inflammation, in turn, involves multiple cell types such as intestinal epithelial cells, innate immune cells as well as adaptive immune cells [4,5]. Meanwhile, in the immune cells, two key transcription factors, Nuclear factor kappaB (NF-κB) and NF-E2p45-related factor 2 (Nrf2), are crucial transcription pathways that regulate a broad range of physiological functions and related genes [3,6,7,8,9]. The products are suggested as either therapeutic targets or biomarkers.

In comparison with the traditional treatment for IBD, nano-drug delivery systems are capable of precisely targeting the inflammatory site, instead of the entire gut. It is beneficial for maintaining long-term remission to cure chronic diseases. Owing to their small size and versatile physiochemical properties, nanomedicines are of particular interest among the accumulation in the inflamed site and response approaches in IBD. Thus, they can effectively enhance the stability of antioxidants and penetrate the antioxidants into inflammatory sites [10,11,12]. In this review, we will focus on the up-to-date antioxidative nanomedicines that have emerged, mainly within the last five years, concerning the management of IBD by oral administration. Depending on the delivery compounds, the antioxidant nanosystems are classified into four catalogs: protein and peptide nanocarriers, nucleic acid nanocarriers, small antioxidant compound nanocarriers, and nanozymes. The last part deals with a general conclusion of concerns in the study and potential research ideas.

## 2. The Reactive Species and Oxidative Stress

The oxidation-reduction reaction is related to all fundamental biological processes [2,3]. The reactive spices are produced by several oxidation processes and can be partially neutralized by the antioxidant defense. In addition to reactive oxygen species (ROS) (e.g., superoxide, peroxides, hydroxyl radical, *α*-oxygen, and singlet oxygen), other types of reactive spices also have remarkable impacts on cellular redox processes, including reactive nitrogen species (RNS) (e.g., nitric oxide and nitrogen dioxide), reactive sulfur species (RSS) (e.g., persulfides, polysulfide, and thiosulfate), and reactive carbonyl species (RCS) (protein aldehydes and protein carbonyls) [1]. The reactive species are generated from both endogenous and exogenous sources. Reactive species produced primarily rely on endogenous enzymatic reactions [2]. The metabolism processes, mitochondrial respiratory chain, prostaglandin synthesis, and phagocytosis are all involved. For instance, myeloperoxidase (MPO), nicotinamide adenine dinucleotide phosphate (NADPH), oxidase, angiotensin II, and lipoxygenase are noticeable [13]. The exogenous sources of reactive species production can occur as a result of exposure to environmental pollutions, heavy metals (e.g., cadmium [Cd], mercury [Hg], lead [Pb], and arsenic [As]), certain drugs (e.g., cyclosporine, tacrolimus, gentamycin, and bleomycin), organic solvents, alcohol, and radiations [14]. Correspondingly, the antioxidant defense from free reactive species’ toxicity can be divided into endogenous and exogenous pathways [2]. Endogenous antioxidants include enzymes, for instance, superoxide dismutase (SOD), catalase (CAT), glutathione peroxidases (GSH-Px), thioredoxin (Trx), and peroxiredoxins (Prxs), as well as the low-molecular-mass antioxidants, such as bilirubin, *β*-carotene, Vitamin E, albumin, and uric acid in plasma. Exogenous antioxidants refer to Vitamin C, Vitamin E, polyphenols, flavonoids, metals (e.g., selenium [Se], copper [Cu], zinc [Zn]), metal oxides, and drugs [1,15,16,17]. Some compounds act as scavengers of reactive species, whereas the others have no such effect directly. The metals or metal oxides are referred to as antioxidant minerals because they defend against oxidative stress by chelation of transition metals and preventing them from catalyzing the production of endogenous reactive species. For instance, Se and Zn have no direct antioxidant function but are necessary for the activity of antioxidant enzymes [18,19,20]. Notably, as has been commonly acknowledged, in comparison to the individual antioxidants, the mixtures exhibit synergistic effects [21,22,23].

Oxidative stress takes place owing to the imbalance between reactive species and antioxidants. It leads to a disorder of redox signaling and damage to biomolecules [24]. The accumulated reactive species, which originate from either endogenous or exogenous sources, cause oxidative modification of the cellular macromolecules: nucleic acids, proteins, lipids, and carbohydrates [2] (Figure 1). The oxidative macromolecules in turn can be employed as biomarkers to quantify oxidative stress [25]. A considerable number of studies demonstrate that oxidative stress has existed in all the aerobic cells and can be responsible, with different degrees of importance, for the onset and/or the progression of common age-related diseases (e.g., cardiovascular disease, cancer, and diabetes) or inflammatory diseases (e.g., metabolic disorders, autoimmune disorders, and IBD) [2,15].

## 3. Oxidative Stress and IBD

IBD is idiopathic chronic and relapsing inflammatory disorder of the gut, which comprises ulcerative colitis and Crohn’s disease [5]. Both are caused by an overactive immune response to gut microbiota in genetically vulnerable individuals [26]. Ulcerative colitis is limited to the colon, whereas Crohn’s disease is regarded as inflammation in the whole gastrointestinal tract in a non-continuous fashion [5,27]. The precise etiology of IBD has been studied for decades and remains unclear. The interaction of various factors, including genetic factors, the immune system, and environmental factors interrupt the homeostasis of the gut (e.g., oxidative stress), leading to inflammatory responses of the intestinal tissue [4,28]. Numerous research pieces of evidence suggest that IBD is associated with the increased production of reactive species. To be exact, multiple studies in colitis animal models proved an augmented formation of reactive species, including superoxide, peroxynitrite, hypochlorous acid, and hydrogen peroxide; meanwhile, the levels of endogenous reactive species-related compounds in colonic tissue, such as glutathione and Cu/ZnSOD, are decreased [28]. Studies using genetically modified animal models to select the appropriate modifications of antioxidant enzymes were the most convincing proof for the cause-and-effect relationship between oxidative stress and IBD [29]. For instance, transgenic overexpressed Cu/ZnSOD significantly attenuated DDS-induced colitis. Depletion of GPx1 and GPx2 or additional glutathione biosynthesis inhibitors (e.g., buthionine sulfoximine) caused the generation of colitis in mice [28,30].

Oxidative stress not only directly damages the intestinal epithelial cells but also causes dysregulated pro-inflammatory reactive species-sensitive pathways in immune cells [7]. The NF-κB signaling and Nrf2 signaling pathways are two key transcription pathways that regulate a broad range of biological functions to respond to oxidative stress and inflammation [9]. The multi-subunit transcription factor NF-κB serves as a pivotal mediator of multiple aspects of both innate and adaptive immune systems. It controls the expression of a series of pro-inflammatory genes, encoding cytokines and chemokines. Additionally, NF-κB directs the survival, migration, and differentiation of immune cells. While reactive species can react with proteins, lipids, polysaccharides, and nucleic acids of NF-κB pathways, this pathway is sensitive to the molecules [6]. In one classic study, NF-κB responded to micromolar concentrations of H_2_O_2_ and this activation was reversed by treatment with antioxidant N-acetyl cysteine (NAC) [31]. Strong evidence indicates that NF-κB is associated with the pathogenesis of IBD patients. The irregulation of NF-κB precursors, NF-κB, the NF-κB stimulating immune receptors (e.g., NOD2), and the down-regulation gens (e.g., interleukins (IL)-12, IL-23) has been found in inflamed colonic tissue of IBD patients [30]. On the other hand, Nrf2 belongs to another family of transcription factors, being capable of inducing a set of antioxidants and detoxication enzymes [3]. The factor-induced transcription of antioxidant proteins is able to protect against the accumulation of overproduced reactive species. The most studied Nrf2-related proteins or protein subunits are NAD(P)H, heme oxygenase-1 (HO-1), dehydrogenase quinone 1 (NQO1), catalytic subunit (GCLC), and the γ-glutamyl cysteine ligase modulatory subunit (GCLM). It is also related to pro-inflammatory cytokines like IL-6, IL-1*β*, and IL-17, extracellular matrix degradation proteins including matrix metalloproteinase (MMPs), and autophagy modulations [32,33]. Additionally, the cellular level of Nrf2 is strictly regulated. For example, the binding with Kelch-like ECH-associated protein1 (Keap1)-Cullin2-Rbx1 complex causes Nrf2 ubiquitination. The stability of the Nrf2/Keap1 complex is sensitive to oxidant stress, while Keap1 protein contains 27 cysteine residues, which can be modified by reactive species [34]. Interestingly, the complex interplay of NF-κB and Nrf2 pathways under conditions of oxidative stress could cause the fine-tuning of dynamic responses by either transcriptional or post-transcriptional mechanisms. For instance, NF-κB directly modulates the Nrf2 transcription and activity, whereas using Nrf2 inhibitor or Nrf2 knockout cells improves the activity of NF-κB leading to increased production of cytokines [35]. Additionally, NF-κB and Nrf2 compete also for the transcriptional co-activator CREB-binding protein (CBP) [9] (Figure 2) .

Above all, oxidative stress plays an important role in the generation and development of IBD. Therefore, besides the conventional methods, targeting the oxidative stress in the intestine by either diminishing the overproduced reactive species or managing antioxidant pathways can effectively treat the disease.

## 4. Antioxidative Nanotherapeutic Approaches for IBD

The classical IBD treatment contains anti-inflammatory drugs (e.g., 5-aminosalicylic acid, glucocorticosteroids), immunosuppressive agents (e.g., azathioprine, 6-mercaptopurine), and anti-tumor necrosis factor (TNF)-*α* monoclonal antibodies (e.g., infliximab, adalimumab) [36,37,38]. Unfortunately, because of non-specific distribution and low retention time, direct administration of the current drugs has the potential to cause side effects and fails to diminish symptoms for a considerable number of patients. As mentioned above, antioxidants, or the compounds targeting oxidative pathways, are capable of balancing oxidative stress and effectively treating IBD. Although experimental and clinical investigations proved the benefits of antioxidants, only limited success has been achieved because of the subsequent challenges. First, the gastrointestinal tract is an enzyme-abundant microenvironment with changeable pH conditions. Thus, the activities of many antioxidants may be significantly compromised. Second, the antioxidants are limited to on-site accumulation [17]. For that reason, many antioxidative NP-mediated strategies have been considered as a remarkably promising platform for IBD treatment. As the physiochemical properties of the NPs (size, surface charge, and surface functionalization) have a strong influence upon their permeation, distribution, and cellular uptake, several targeting strategies to design nanocarriers for IBD treatment are employed [39]. The size-dependent accumulation of NPs is mostly studied. To be more concrete, since intestinal inflammation induces the enlarging of tight junctions and increasing permeability, NPs with appropriate sizes can passively accumulate at the inflammatory site [40]. The surface charges are another important character that needs to be tuned for the NPs designed for IBD treatment. The optimal surface charges for them are negative, because the targeting inflammation of the colonic mucous membrane is accumulated of positively charged proteins [41]. Another welcomed delivery strategy is to develop NPs responses to high levels of reactive species at the intestinal inflammatory area [42]. On the other hand, chemical and molecular mechanisms of the delivered antioxidants also have a strong impact on the therapeutic effect. In the following sections, the IBD nanomedicines will be introduced and summarized according to the classification of transported antioxidants (Table 1) (Figure 3).

### 4.1. Nanosystem Delivery of Protein and Peptide Drugs to Impact Oxidative Stress

Oral administration of proteins and functional peptides is particularly challenging for therapeutic approaches to IBD treatment because of their instability in the gastrointestinal tract. Nevertheless, some researchers reported nanoplatforms to encapsulate poor soluble proteins, which are native antioxidant enzymes, naturally derived products with antioxidant activities, or immune system-specific targeting antibodies [22,43,44,45,46,47]. Zen et al. reported SOD and CAT can be directly one-step co-loaded in the nanoparticles and self-assembled by amphiphilic wind chimes like cyclodextrin (WCC) in an aqueous solution under physiological conditions. SOD/CAT co-loaded WCC NPs could not only maintain the activity of endogenic SOD and CAT but also effectively promote the cellular uptake of exogenous antioxidant enzymes. Consequently, the ability to scavenge reactive species, produced by lipopolysaccharide treated macrophages, was increased. The secretion of inflammatory factors decreased, indicating inflammation was inhibited [43]. Another economic protein NP product has been investigated as a carrier for this application as well. In 2019, Huang and coworkers designed NPs by associating chitosan with fucoidan, an anionic long chain sulfated polysaccharide obtained from brown algae for targeting the delivery of soluble eggshell membrane protein (SEP). SEP is extracted from egg products and shows antioxidant and anti-inflammatory activities in intestinal tissues. The chitosan and fucoidan-formed NPs protected the protein from acidic degradation and controlled its release by the response to pH variation in the gastric intestinal tract. Furthermore, the antioxidant activities of encapsulated SEP were significantly enhanced [43]. Administration of TNF-*α* antibodies is another promising class of drugs owing to enormous achievement in the treatment of inflammatory diseases [41,44]. TNF-*α* plays a crucial role in IBD, since it is the main pro-inflammatory cytokine primarily secreted by macrophages further targeting the mitochondrial metabolism and leading to an augmented consequence during IBD [77]. However, the immunosuppression caused by systemic exposure to antibodies leads to adverse effects as well as low efficiency. In order to improve antibody therapy for IBD, Yang and coworkers recently demonstrated a nano-platform to orally deliver TNF-*α* antibody, infliximab, by using hydrogen bonding supramolecular NPs assembled with tannic acid and 1,2-distearoy-sn-glycero-3-phsphoethanolamine-N-[methoxy(polyethylene glycol)] (DSPE-PEG). In this way, Infliximab was protected in the intestinal tract without denaturation/degradation and targeted the intestinal inflammatory site with a high level of reactive species. Thus, a significantly increased therapeutic strategy compared to unprotected antibodies was achieved [44].

In comparison with the proteins, peptides have a smaller molecular size and better solubility in the physiological aqueous environment. The encapsulation of peptides into NPs could ensure that the peptides are more stable and effective due to targeted delivery and sustained release. For instance, naturally occurring tripeptide KPV (Lys-Pro-Val), derived from *α*-melanocyte-stimulating hormone (MSH), shows anti-inflammatory effect and antioxidative properties on treating colitis. However, the tripeptide is not stable in the intestinal environment without protection. For that reason, Xiao et al. fabricated KPV-loaded hyaluronic acid (HA)-functionalized PLGA NPs with a negative surface charge and desirable size (approximately 270 nm). NPs were biocompatible with intestinal cells and accelerated mucosal healing by attenuating inflammation. The NPs were further loaded in the chitosan/alginate hydrogel system. The HA-KPV NPs encapsulated chitosan/alginate hydrogel system displayed a strong capacity to protect mucosa and down-regulate TNF-*α*. The results demonstrated that the nano-in-gel system can long-term release HA-KPV NPs in the colon. Then the NPs penetrated colitis tissues and enabled antioxidative tripeptide internalization to alleviate inflammation [47].

### 4.2. Nanosystem Delivery of Nucleic Acid Drugs to Interfere with Antioxidant Pathways

The development of another macromolecule, nucleic acids’, delivery nanomaterials is attracting great attention in antioxidant therapy. The nucleic acids-mediated antioxidative nanotechnology has the potential to precisely inhibit oxidative stress-induced molecular damages, meanwhile, unexpected interference can be avoided [78]. The nanocarriers are typically designed for oral administration, which is the most appropriate and cost-effective approach to deliver encapsulated nucleic acid to gastrointestinal tissues [43,79]. Therapeutic nucleic acids such as messenger ribonucleic acids (mRNAs), micro ribonucleic acids (miRNAs), and small interfering ribonucleic acids (siRNAs) can be delivered by nanocarriers to regulate oxidative stress-related genes for IBD treatment. Since the physicochemical properties of polymers can be carefully tuned, the utility of functional polymers as intracellular delivery systems for nucleic acids has been wildly used in clinical trials [80]. Various polymers such as modified nature-derived polymers, amphiphilic copolymers, and siRNA-polymer conjugates can condense the nucleic acids (negatively charged and hydrophilic) into the carriers via electrostatic interactions and hydrophobic interactions [79].

In 2018, the modified mRNA molecule for expressing a desired anti-inflammatory cytokine (e.g., IL-10) was developed by Dan Peer and his coworkers to effectively treat IBD. The modified mRNA-loaded lipid NPs were mainly formed by distearoylphosphatidylcholine(DSPC), cholesterol, 1,2-Dimyristoyl-rac-glycero-3-methoxy(DMG)-PEG and 1,2-distearoyl-sn-glycero-3-phosphorylethanolamine(DSPE)-PEG. In order to precisely target Ly6c+ inflammatory leukocytes, the NPs were further functionalized by anti-Ly6c monoclonal antibodies [48].

MiRNAs are endogenous single-stranded non-coding RNAs (approximately 22 nucleotides). They act on post-transcriptional regulators of gene expression [81,82]. Recently, miRNAs have been found to regulate responses to oxidative stress. A significant number of publications have described the targeting of miRNAs on the Nrf2 pathways or GSH biosynthetic enzymes [83,84]. Therefore, the delivery of miRNAs or synthetic miRNA inhibitors by nanotechnology can be promising medical treatments of IBD. For instance, miRNA-31 has been found to be elevated in colon tissues from both Crohn’s patients and colitis patients. MiRNA-31 inhibitors or miRNA-31 inhibitors/curcumin encapsulated *α*-lactalbumin NPs in Konjac glucomannan (sOKGM) microspheres successfully reduced features of colitis and further treated colorectal cancer [49].

Similar to miRNAs, siRNAs also have the potential to treat wide-ranging classes of diseases, because they are capable of reversibly silencing target genes [85]. Several studies have documented that targeted siRNA nanocarriers ensured the penetration of siRNA from the surface of inflamed tissue into the immune system. siRNAs could target macrophage cells, directing against pro-inflammatory cytokines (e.g., TNF-*α*) and cytokines related kinase (e.g., mitogen-activated kinase kinase kinase kinase 4 abbreviated as Map4k4) to treat intestinal inflammatory diseases [41,50,51,86,87]. Given the important role of TNF-*α* in IBD progression, Murthy and his co-workers operated a thioketal delivery system, which locally released TNF-*α* siRNA in response to reactive species at the site of inflammation to treat DSS-induced colitis in mice. The nanomaterial was formed from a polymer, poly-(1,4-phenyleneacetone dimethylene thioketal) (PPADT), which enabled the protection of siRNA from the harsh environment. Most importantly, PPADT NPs degraded selectively in response to reactive species. Taken together, the TNF-*α* siRNA-loaded PPADT NPs effectively silenced TNF-*α* expression in mice suffering from colitis [88]. Since then, the smart reactive species-response polymer has made a significant contribution to the treatment of numerous gastrointestinal inflammatory diseases such as IBD and gastrointestinal cancers. In the subsequent studies, galactosylated low molecular weight chitosan (gal-LMWC), mannosylated poly (amido amine)/sodium triphosphate (TPP), calcium phosphate (CaP)/poly (lactic acid-co-glycolic acid) (PLGA), poly(ethylene glycol)-block-poly(lactic-co-glycolic acid)(PEG-b-PLGA)/cholesterol, poly(lactic-co-glycolic acid)(PEG-b-PLGA)/galactosylated chitosan and poly(ethylene glycol)-b-poly(trimethylene carbonate-co-dithiolane trimethylene carbonate)-b-polyethylenimine (PEG-P(TMC-DTG-PEI)) triblock copolymer have also been reported in the literature as TNF-*α* siRNA carriers to treat colonic inflammatory diseases [25,41,50,51,89,90]. The TNF-*α* siRNA-loaded NPs were also used as a matrix for the co-delivery of inflammatory drugs such as dexamethasone sodium phosphate (DXMS) and curcumin [50]. Studies also showed that the regulation of Map4k4 not only mediated TNF-*α* signaling but also promoted its expression. Upon oral administration, the Map4k4 siRNA, encapsulated in galactosylated trimethyl chitosan-cysteine (GTC)/tripolyphosphate (TPP) or GTC/HA NPs, significantly decreased the expression of TNF-*α* in colonic cells and related parameters in the DSS-induced colitis in a mouse model [91,92].

### 4.3. Nanosystem Delivery of Small-Molecule Antioxidants to Act as Reactive Species Scavengers

Both natural and synthetic small-molecule antioxidants have been widely studied for IBD treatment. The commonly used natural antioxidants include bilirubin, polyphenols, flavonoids, genipin, glutathione, etc., whereas synthetic antioxidants are edaravone, lipoic acid, NAC, 4-Hydroxy-2,2,6,6-tetramethylpiperidinyloxyl (Tempol), etc. [17].

Given that naturally derived antioxidants are vulnerable, researchers have attempted to develop naturally derived antioxidants delivery nanomaterials. As a result, antioxidants can be effectively delivered to specific inflammatory intestinal sites [78]. For example, bilirubin has been suggested as a potent endogenous antioxidant that is capable of scavenging reactive species to protect cells/tissues from oxidative damage. Although the physiological role of bilirubin has been investigated for decades, the clinical application of bilirubin has been restricted due to its poor water solubility [71,93,94,95]. To overcome the critical problem of natural bilirubin, Jon and his coworker worked on PEGylated bilirubin NPs. The self-assembled particles, with a diameter of approximately 110 nm, are highly efficient hydrogen peroxide scavengers, which protect cells from hydrogen peroxide-induced damage. In vivo, the PEGylated bilirubin NPs showed preferential accumulation at the inflammatory site and significantly inhibited the inflammatory response in the colon [94,95]. Moreover, another bilirubin-derived NP, HA-bilirubin, has been developed. The NPs accumulated in the inflamed colonic epithelium in mice and have multiple positive effects including restoring epithelium barriers, augmenting the overall abundance of gut microbiota, and effectively regulating innate immune responses [71].

Moreover, polyphenols are a large family of phytochemicals that are abundant in food and derived from plants characterized by multiples of phenol units. However, their drawbacks such as intrinsic poor water solubility and low bioavailability after generally oral administration need to be overcome. The utility of polyphenol in nano-formulations aims to improve its solubility, activity, and stability, making the compound therapeutically more effective without adverse effects. Many nano-encapsulated polyphenols (curcumin, resveratrol, berberin, epigallocatechin gallate (EGCG), tannic acid, rosmarin acid, oleuropein, and ginsenoside) have recently been proved to have anti-inflammatory properties and have an important role in the management of IBD [10,52,53,54,55,56,57,58,59,60,61,62,67]. For the same reason, the polyphenol-rich extracts such as grape seed extract, green tea extract, and lycin barbarum extract have the potential to treat intrinsic inflammation as well [66,68,69,70]. Currently, a compound used worldwide is curcumin, which is derived from Curcuma longa extracts. Regarding its antioxidant and anti-inflammatory effects, multiple treatments have been remarkably highlighted [96]. To overcome the drawbacks of curcumin, a novel fibroin/chitosan-based macrophage-targeted curcumin delivery system was developed by Gou et al. The fibroin/chitosan NPs have well-controlled size distribution (approximately 175.4 nm), negative surface charge, and effective curcumin encapsulating. Upon the stimulation by pH/GSH/reactive species, curcumin can be controlled released. Due to the surface characteristics of the NPs, they can specifically recognize and bind to the glycoprotein CD44 on the surface of macrophages. As a result, the cellular uptake capacity of NPs is improved through the CD44 mediated endocytosis pathway. Through both oral administration and intravenous therapy, the particles could improve the specific internalization and exhibit controlled release of the compound [58]. In addition, HA-functionalized chitosan/PLGA, hydrophilic Eudragit^®^ S100, hydroxyethyl starch-curcumin conjugates, genipin-crosslinked human serum albumin, chitosan/sodium alginate/cellulose acetate phthalate polyelectrolyte multilayer, and *α*-lactalbumin/sOKGM have been recently explored to encapsulate or co-encapsulate curcumin for IBD treatment [56,57,59,87,97,98].

Accumulating studies have reported that flavonoids (e.g., quercetin, catechin, silymarin) showed beneficial effects in treating IBD [63,64,66]. The reasons for the powerful effects are first, that they can act as strong antioxidants, and second, that they can act as cellular modulators of protein kinase and lipid kinase signaling pathways. [99] Most recently, genistein has been delivered by *β*-cyclodextrin(*β*-CD) and 4-(hydroxymethyl)phenylboronic acid pinacol ester-modified genistein nanosystems (defined as Gen-NP2). Gen-NPs could effectively release genistein to the inflamed colon instead of absorption by the stomach or intestines. Gen-NPs effectively scavenged reactive species and regulate the inflammasome-autophagy pathway. Spontaneously, gut microbiota were modulated. Eventually, intestinal mucosal healing and barrier integrity were promoted [65].

On the other hand, synthetic antioxidant compounds can also be potentially useful in IBD therapy. The synthetic antioxidant-loaded NPs not only have enzyme-mimicking functions but also spontaneously scavenge reactive species during the catalyzing [17]. Thus, they have been engineered as another type of candidate for IBD treatment. Nagasaki designed a ROS-nitroxide radical-containing particle (RNP_o_), having a diameter of 40 nm, by establishing an amphiphilic block copolymer with Tempol, which is a stable nitroxide radical-containing ROS trapper. They could specifically accumulate in the IBD model. In comparison with Tempol and 5-aminosalicyclic acid, RNP_o_s were more effective in reducing inflammation. The RNP_o_s could further load silymarin, an active compound with anti-inflammatory and antioxidant properties, resulting in a synergic effect for the recovery in the colonic mucosa of the DSS-induced model [100]. A different approach was followed by Zhang and coworker. They fabricated a series of SOD/CAT-mimetic nanomedicine based on PBAP-conjugated *β*-CD material. Tempol and annexin A1-mimetic peptide Ac2-26 were effectively packed into smart-responsive nanocarriers. Benefiting from the protection and site-specific accumulation of the nanomaterial, both Tempol and Ac2-26 were control released at the inflammatory sites. Owing to the therapy by RBAP-conjugated *β*-CD-based nanomaterial, the inflammatory symptoms were reduced, the wound healing of intestinal mucosal accelerated, and the composition of gut microbiota reshaped [45]. Another method is to develop a nanoscale prodrug. IBD targeting Janus-prodrug (Bud-ATK-Tem) was conjugated by the anti-inflammatory drug budesonide (Bud) conjugated ROS-responsive aromatized thioketal (ATK) and Tempol. Due to macromolecular interaction, hydrophobic interactions, and π-π interaction of the amphiphilic conjugate, the prodrug self-assembled into NPs with the size of approximately 100–120 nm. The 98% drugs (Bud and ATK) were released in the inflammatory macrophages. In the DSS colitis model, the drug-loaded NPs were passively accumulated in the inflamed tissue, thus ensuring they can improve the antioxidative and anti-inflammatory efficacy [101].

### 4.4. Nanozymes to Catalyze Oxidative Defense

Some specific metals and metal oxides have inherent enzymatic properties which have been known for decades. In 2007, Yan and coworkers investigated iron oxide formed peroxidase mimic nanomaterials. Since then, metal-based antioxidative nanomaterials were defined as nanozymes [102]. The new generation of artificial enzymes not only has the advantages of unique properties of nanomaterials but also exhibits high catalytic activity, superior stability, and economical price, among others. Therefore, various nanomaterials formed with metals and metal oxides have been rapidly studied and industrialized for therapeutic applications [103]. As mentioned above, Se represents the most significant part of the active center of antioxidant enzymatic activities (selenoproteins) [104]. To date, various Se-based compounds have attracted great attention due to their inherent antioxidant enzyme-like property. Se-NPs produced by L.casei ATCC 393 significantly alleviated the increase of reactive species and maintained permeability of intestinal epithelial cells (NCM460 cell line). Particularly, the Se-derived NPs diminished the ultrastructure damage of mitochondria caused by oxidant stress [72]. Besides the above-mentioned nanozymes, some other inorganic NPs have also been investigated, such as NPs derived from Prussian blue, manganeses (Mn), Cerium oxide (CeO_2_), ZnO, and Gold (Au) [11,18,19,20,50,73,74,75,76]. A remarkable work was reported by Chen and coworkers, who synthesized Mn-Prussian blue NPs with multi-enzyme-like properties to mediate catalytic IBD therapy. Owing to the positively charged artificial surfaces and desired sizes, NPs significantly improved colitis in mice via the toll-like receptor (TLR) signaling pathway with no adverse effects [18]. Another notable versatile nanozyme for biological application was CeO_2_ NP. Upon the mixed-valence states between Ce^3+^ and Ce^4+^ on the surface of NPs, it has been demonstrated to possess SOD and CAT-mimetic activity as well as the activities of scavenging reactive species. Due to the presence of surface oxygen vacancies, Ce^4+^ can be reduced to Ce^3+^, resulting in effectively decreasing ROS levels [105]. An in situ growth NPs was performed by Zhao and coworkers. They reported a system for IBD treatment coupled with multi-enzyme mimicking CeO_2_ NPs and montmorillonite (MMT). When the CeO_2_/MMT ratio was 1:9, the nanozyme was stable in the gastric tract by oral administration. The NPs were more effective and stable than free enzymes. Moreover, upon electrostatic interactions, negatively charged MMT was associated with specific targeting to positively charged inflamed colon tissue [19].

## 5. Conclusions and Prospects

To date, IBD has evolved into a worldwide disease in not only developed countries but also newly industrialized countries [106]. Accumulated studies suggest that IBD is caused by commensal microbe-induced continuing inflammation in a genetically vulnerable host [26]. During the inflammatory process, the inflammatory cells related to the immune system secrete a large number of cytokines and chemokines, which stimulate reactive species overproduction and eventually cause oxidative stress [5]. Given oxidative stress plays a crucial role in the pathogenesis of IBD, multiple antioxidant therapeutic strategies are being explored including the removal of reactive species, enhancing the synthesis of antioxidant enzymes, mimicking the antioxidant enzymes, and the inhibition of abnormal redox signaling for reactions.

Engineering materials with either naturally derived polymers or synthetic polymers at the nanoscale enables the arrangement of several above-mentioned strategies into one delivery system with multicomponent and multifunction. In other words, using nanotechnology can spontaneously co-carry antioxidant pathways related macromolecules (proteins, peptides, and nucleic acids) and small molecules (antioxidants and metal oxides) to treat IBD. Meanwhile, many benefits can be achieved, including improving the delivery of poorly water-soluble antioxidants, targeting delivered drugs in an inflammatory manner, and transcytosis of functional compounds across the epithelial barriers to immune cells [39]. However, to successfully translate current nanotherapeutic approaches from laboratory investigation to clinical application, numerous limitations still need to be overcome. In general, minimizing the batch-to-batch variation, improving pharmacokinetic properties, enhancing loading efficiency, handling cost-efficiency, and simplifying synthesis progress are key points.

There are several concerns and challenges associated with antioxidant monotherapy for IBD treatment. First, oxidative stress might not be the primary contributor to disease. In other words, avoiding overproduced reactive species may not have a key impact on the progression of inflammatory diseases. Thus, future experimental and clinical studies should not only evaluate the therapeutic efficacy of the antioxidants (macromolecules or small molecules) alone but also study the synergic effects of antioxidants, combined with conventional drugs and engineered stem cells (e.g., allogeneic expanded adipose-derived mesenchymal stem cells). Second, complex pathophysiological mechanisms should be taken into consideration. Systematic exploration of protein complex composition and network analysis between NF-κB and Nrf2 pathways can provide new insight to manipulate IBD [9]. To our knowledge, the crosstalk of Nrf2 and NF-κB has not been systematically investigated. In the coming future, researchers can focus on the direction of the crosstalk between the NF-κB and Nrf2 pathways by nanotechnology. Third, future studies should also define the correlation between the endogenous attenuation of oxidative stress and exogenous antioxidant treatment. It should always be kept in mind that reactive species have important physiological functions, especially in immunity against pathogenic microorganisms. The ideal antioxidant-based therapeutic approaches should be designed and evaluated precisely to decrease oxidative damage without significantly diminishing the influence of reactive species on physiological activities. Last but not the least, researchers can also cooperate with the diagnostic materials (e.g., graphene quantum dot) in addition to the antioxidant nanomaterials [107]. Thus, multifunctional antioxidative nanomedicines can monitor the pharmacokinetics and pathological process for designing accordingly personalized antioxidant therapy.

## Figures and Tables

**Figure 1 biomedicines-10-00085-f001:**
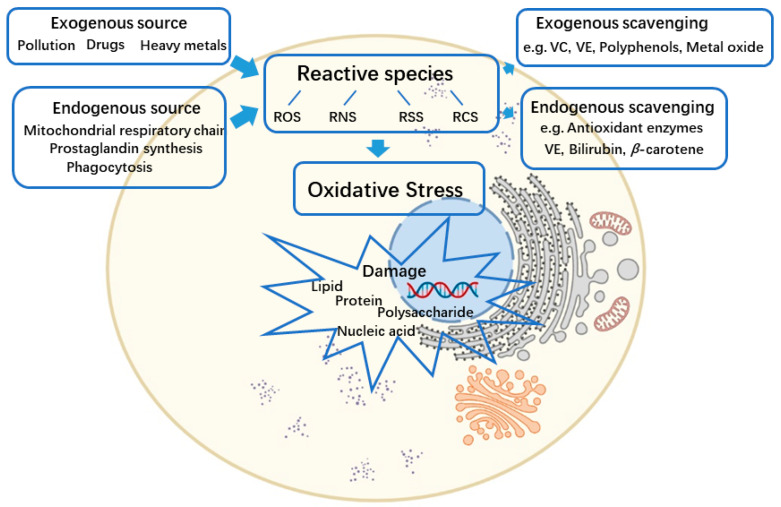
The imbalance between the generation of reactive species and the antioxidant defense system results in oxidative stress and further damage to cellular macromolecules. ROS—reactive oxygen species; RNS—reactive nitrogen species; RSS—reactive sulfur species; RCS—reactive carbonyl species; VC—ascorbic acid; VE—tocopherol. Created by BioRender.com.

**Figure 2 biomedicines-10-00085-f002:**
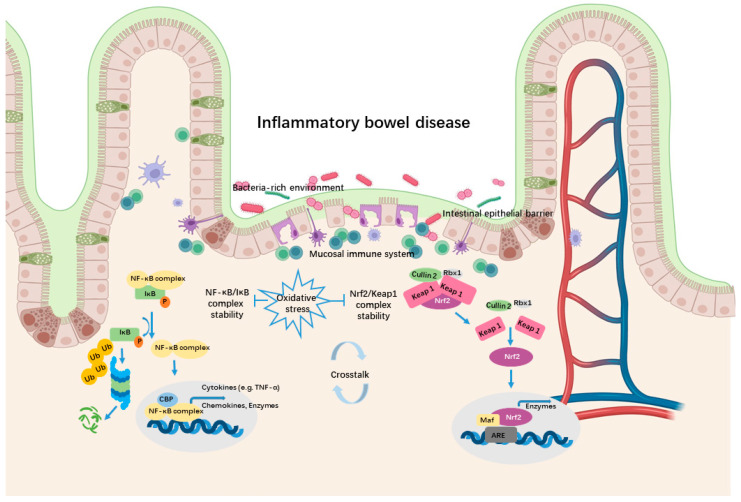
Schematic representation of the inflammatory response of antioxidant pathways in the intestinal environment. In the immune cells, oxidation stress can enhance the dissociation of the NF-κB/IκB complex and the Nrf2/Keap1 complex, which cause the induction of pro-inflammation genes (e.g., cytokines, chemokines) and antioxidant genes (e.g., enzymes). The crosstalk between these two pathways through a complex molecular interaction plays an important role in IBD. NF-κB—nuclear factor-kappaB; Ub—ubiquitination; CBP—CREB-binding protein; Nrf2—NF-E2p45-related factor 2; Keap1—Kelch-like ECH-associated protein 1; ARE—antioxidant responsive element. Created by BioRender.com.

**Figure 3 biomedicines-10-00085-f003:**
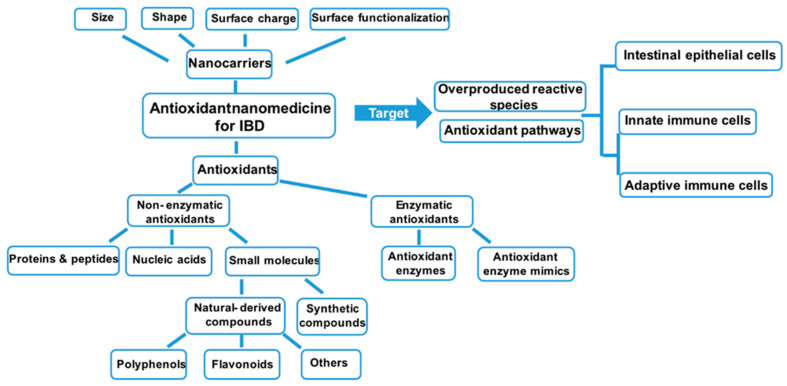
Multiple antioxidant nanomedicines are designed to scavenge the overproduced reactive species (non-enzymatic antioxidants) or enhance the catalyzation of antioxidant processes (enzymatic antioxidants), leading to attenuate the inflammation within the gut. In order to specifically target the inflammatory site (intestinal epithelial cells or the intestinal immune system), the size, shape, surface charge, and surface functionalization should be taken into consideration. The classification of the generally-used antioxidants to treat IBD has been summarized. IBD—inflammatory bowel disease.

**Table 1 biomedicines-10-00085-t001:** Examples of new innovative antioxidant nanotherapeutic approaches against IBD within the last 5 years.

Antioxidants	Type of Compounds	Nanosystem Components	Size and Surface Charge	Colitis Model	References
**Protein/peptide**					
SOD/CAT	Antioxidante enzyme	WCC	~156 nm	DSS-induced mice	[43]
TNF-*α* antibody/tannic acid/ EGCG	Protein/polyphenol	DSPE-PEG	~100 nm	DSS-induced mice	[44]
Ac2-26	Peptide	PBAP conjugated*β*-CD	202 ± 4 nm, −37.4 ± 0.6 mV	DSS-induced mice	[45]
SEP	Protein	Chitosan/Fucoidan	tunable	LPS-induced macrophage	[46]
Anti-TNF-*α* antibody	Protein	Galactose/PLGA	~261 nm, ~−6 mV	DSS-induced mice	[22]
KPV	Peptide	PLGA/PVA/HA/chitosan	~270 nm, −5.3 mV	DSS-induced mice	[47]
**Nucleic acid**					
IL-10 mRNA	Modified mRNA	Lipid	63.7 ± 1.59 nm	DSS-induced mice	[48]
Anti-miRNA-31/Curcumin	MiRNA inhibitor/polyphenol	*α*-lactalbumin/OKGM (nano-in-micro)	~25 μm, ~−7 mV	AOM-DSS-induced	[49]
TNF-*α* siRNA/Dexamethasone	SiRNA/small molecule	TKPR-PEG-P(TMC-DTC), PEG-P(TMC-DTC)-PEI	~500 nm, ~0.6 mV	DSS-induced mice	[50]
TNF-*α* siRNA	SiRNA	PEG-b-PLGA	~120 nm, −17 mV~31 mV	DSS-induced mice	[41]
TNF-*α* siRNA	SiRNA	PVA/PLGA	~300 nm, ~20 mV	DSS-induced mice	[51]
**Small molecule**					
Curcumin/Dex	Polyphenol/ glucocorticoid	PLGA/HPMCAS-HF (nano-in-micro)	~176 nm	HT29-MTX/T84 cell line	[52]
Curcumin	Polyphenol	Chitosan/alginate/cellulose	421 ± 14 nm, −47 ± 3 mV	DSS-induced mice	[53]
Curcumin/tannic acid	Polyphenol	Genipin-crosslinked HBA	~220 nm, −28.8 mV	TNBS-induced mice	[54]
Curcumin	Polyphenol	Silk fibroin/Chondroitin sulfate	~175.4 nm, −35.5 mV	DSS-induced mice	[55]
Curcumin	Polyphenol	Eudragit® S100		DSS-induced mice	[56]
Resveratrol	Polyphenol	*β*-Lactoglobulin	165 ± 2 nm, −34 ± 0.6 mV	Winnie mice	[57]
Resveratrol	Polyphenol	PLGA/chitosan/alginate	255.9 ± 12.0 nm, 13.5 ± 3.9 mV	DSS-induced mice	[58]
Resveratrol	Polyphenol	Chitosan/pHEMA/in pDMAEMA (nano-in-gel)	121 ± 1 nm, −170 ± 90 mV	DSS-induced mice	[59]
Rosmarinic acid	Polyphenol	Chitosan/nutriose	63.5 ± 4.0 nm, −33.70 mV	DSS-induced mice	[60]
Rosmarinic acid	Polyphenol	PEG	141.2 ± 12.3 nm, −25.30 ± 2.7 mV	DSS-induced mice	[10]
Oleuropein	Polyphenol	Lipid	~ 150 nm, −25 mV	DSS-induced mice	[61]
EGCG	Polyphenol	Amyloid	-	DSS-induced mice	[62]
Tannicacid/EGCG/catechin	Polyphenol/glucocorticoid	Block PEG	~130 nm, −27 mV	DSS-induced mice	[63]
Quercetin	Flavonoids	Silk fibroin	175.8 ± 0.9 nm, −24.5 ± 4.1 mV	DSS-induced mice	[64]
Genistein/Tempol/VE	Flavonoids/	*β*-CD/HMPBA/TPGS	636 ± 94 nm/304 ± 60 nm	DSS-induced mice	[65]
Silymarin	Synthetic antioxidant compound	Silica-derived	−21.08 ± 1.51/6.63 ± 1.91 mV	DSS-induced mice	[66]
Ginsenoside	Flavonoids	Glycogen-derived	~110 nm	DSS-induced mice	[67]
Grape seed extract/	Steroid glycosides	Grape seed extract/	128.9 ± 0.3 nm, 1.3 ± 0.08 mV	DSS-induced mice	[68]
Horseradish peroxidase	Plant extract/antioxidant enzyme	Horseradish peroxidase			
Lycium barbarum	Plant extract	Lipid	~189.2 nm, ~−34.9 mV	DSS-induced mice	[69]
Green tea extract	Plant extract	PLA-PEG	~163.1 nm, ~−7.92 mV	TNBS-induced rat	[70]
Bilirubin	Small molecule	HA	86 ± 5 nm to 416 ± 9 nm	DSS-induced mice	[71]
			−35.6 ± 1.6 mV to −46.2 ± 5.2 mV		
**Nanozyme**					
CeO_2_	Nanozyme	Red blood vesicle/exosome	~3 nm	DSS-induced mice	[11]
CeO_2_	Nanozyme	MMT/CeO_2_	1.6 ± 0.2 nm, −30.3 ± 0.3 mV	DSS-induced mice	[19]
Prussian blue/Mn	Nanozyme	PVP	60 nm~120 nm, −27.0 mV	DSS-induced mice	[18]
Prussian blue	Nanozyme	PVP	~60 nm	DSS-induced mice	[20]
Se	Nonozyme	Lactobacillus casei produced	50~80 nm	NCM460 cells	[72]
Se	Nanozyme	Enterobacter cloacae Z0206 produced	139.43 ± 7.44 nm	DSS-induced mice	[73]
Se	Nanozyme	Ulva lactuca polysaccharide	30 to 150 nm	DSS-induced mice	[74]
Gold	Nanozyme	PVP/Citrate	~5 nm	DSS-induced mice	[75]
ZnO	Nanozyme	ZnO	29.7 ± 4.0 nm, −59.4 ± 3.8 mV	DSS-induced mice	[76]

## Data Availability

No applicable.

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
