# Peer review of "Oxidative Stress and Antioxidant Nanotherapeutic Approaches for Inflammatory Bowel Disease"

_biomedicines, 2021, doi:10.3390/biomedicines10010085_

Round 1

Reviewer 1 Report

This manuscript reviews an interesting topic but some comments must be addressed.

  • The Authors described oxidative stress “as a result of accumulation of reactive species interactome (RSI)” (page 1 line 12 and 24-25; page 3 line 83) and RSI as “RSI is produced by several oxidation processes” (page 2 line 55). However, RSI is related to the physiological functions of reactive species and their chemical interactions and not the reactive species alone, as defined in this manuscript. In my opinion, the RSI should be replaced by reactive species (or ROS/RNS or RONS), and the corrected definition of RSI should be included.
  • In Figure 1, VC and VE abbreviations should be written out in full on figure legend.
  • Figure 3 quality should be improved.
  • On page 8, line 311, “in vivo” is not italicized.
  • In Table 1 font size is too small, and the table is cited on page 5 but was included on page 10.

Author Response

29/12/2021
Re: Submitted Manuscript Biomedicines-1516050
Dear editor and reviewer,
We would like to express our appreciation for the detailed and careful review of our manuscript entitled “Oxidative stress and antioxidant nanotherapeutic approaches for inflammatory bowel disease”.
Consideration of the issues raised by the reviews has undoubtedly strengthened the manuscript. We have addressed every point raised by the reviewers and have answered accordingly and modified the manuscript accordingly. The changes are marked up using the “track changes” function in the revised manuscript. Our responses to the comments are provided in red.
Point 1: The authors described oxidative stress “as a result of accumulation of reactive species interactome (RSI)” (page 1 line 12 and 24-25; page 3 line 83) and RSI as “RSI is produced by several oxidation processes” (page 2 line 55). However, RSI is related to the physiological functions of reactive species and their chemical interactions and not the reactive species alone, as defined in this manuscript. In my opinion, the RSI should be replaced by reactive species (or ROS/RNS or RONS), and the corrected definition of RSI should be included.
Response 1: Thanks for your valuable comments. According to your suggestion, we have replaced the reactive species interactome(RSI) with reactive species. The correct definition of RSI is the chemical interactions of reactive species among themselves and with downstream biological targets [1]. In our manuscript, we focus a bit more on the species alone instead of the whole system.
Point 2: In Figure 1, VC and VE abbreviations should be written out in full on figure legend.
Response 2: Thanks. The abbreviations have been written out in full, on figure legend.
Point 3: Figure 3 quality should be improved.
Response 3: We improved the quality of Figure 3 as suggested.
Point 4: On page 8, line 311, “in vivo” is not italicized.
Response 4: Thanks, we changed the font style of this word (page 9, line 324).

Reviewer 2 Report

In this manuscript Liu et all. gives an overview about nanotherapeutic approaches for the treatment of inflammatory bowel disease with special focus on antioxidant delivery. They give a nice overview about the possibilities in the field and some examples. However, there are some points which need to be improved:

General:

  • Figure 1: Please add abbreviations for VC, VE, RSI
  • Figure 2: Please give examples in the figure legends for pro-oxidants/pro-inflammation genes (mention the ones shown in the figure). In addition, in the figure it is shown that Nrf2 induce antioxidant proteins, but it is not describes in the legend, please clarify.
  • Figure 3: Figure legend is to general and not specific enough for the Figure please clarify. In addition, please add abbreviations (RSI, IBD).
  • Paragraph 4.2: Please include also examples for the delivery of mRNA in this section
  • Paragraph 5: Conclusion is to general and mainly discusses the challenge of an antioxidant therapy. Please go into detail about nanotherapeutic approaches in the conclusion
  • Please include a paragraph about the limitations of nanotherapeutic approaches.

Author Response

29/12/2021
Re: Submitted Manuscript Biomedicines-1516050
Dear editor and reviewer,
We would like to express our appreciation for the detailed and careful review of our manuscript entitled “Oxidative stress and antioxidant nanotherapeutic approaches for inflammatory bowel disease”.
Consideration of the issues raised by the reviews has undoubtedly strengthened the manuscript. We have addressed every point raised by the reviewers and have answered accordingly and modified the manuscript accordingly. The changes are marked up using the “track changes” function in the revised manuscript. Our responses to the comments are provided in red.
Point 1: Figure 1: Please add abbreviations for VC, VE, RSI
Response 1: Thank you. According to your suggestion, we added the full words of the abbreviations.
Point 2: Figure 2: Please give examples in the figure legends for pro-oxidants/pro-inflammation genes (mention the ones shown in the figure). In addition, in the figure it is shown that Nrf2 induce antioxidant proteins, but it is not described in the legend, please clarify.
Response 2: Thanks for your valuable comment, we revised our figure and figure legend accordingly. As we described in the manuscript, the Nrf2 is capable of inducing a set of antioxidant-related enzymes. We replaced the antioxidant proteins with enzymes to make the figure more suitable. And, we also added examples in the figure legend.
Point 3: Figure 3: Figure legend is to general and not specific enough for the Figure, please clarify. In addition, please add abbreviations (RSI, IBD).
Response 3: Thanks. As suggested, we added more details in the figure legend. Moreover, we added the full words of the abbreviation.
Point 4: Paragraph 4.2: Please include also examples for the delivery of mRNA in this section.
Response 4: Many thanks for your advice. Recently, mRNA delivery has increased the attention, while it has broad potential as a therapeutic approach focused on vaccination,